# Determining the ideal length of wind speed series for wind speed distribution and resource assessment

Lihong Zhou[1], Igor Esau[1]

[1]Department of Physics and Technology, Faculty of Science and Technology, The Arctic University of Norway, Tromsø, 9010, Norway

*Correspondence to*: Lihong Zhou (lihong.zhou@uit.no)

**Abstract.** Accurate wind resource assessment depends on wind speed data that capture local wind conditions, which are crucial for energy yield estimates and site selection. While the International Electrotechnical Commission (IEC) recommends at least one year of data collection, this duration may be insufficient to fully account for interannual variability. Although studies often maximize data length, limited guidance exists on the minimum duration required to reliably estimate wind statistics and energy potential. To address this gap, we propose a method to quantify the errors in wind speed distribution parameters introduced by using time series of varying lengths, compared to long-term reference data. This enables us to determine the minimum number of hourly observations needed to achieve a given accuracy. We apply this method to both in-situ weather station observations and ERA5 reanalysis data at 10 m and 100 m heights. Our results show that basic parameters, such as mean, standard deviation, and Weibull parameters, can stabilize with ~1 month of hourly data, while higher-order moments such as skewness and kurtosis require substantially longer records ($\geq$ 1.6 years and 88.6 years, respectively). Although ERA5 stabilizes faster, it exhibits systematic biases compared to in-situ measurements. Moreover, random sampling (combining available hourly data) can yield comparable distribution parameters to diurnally or seasonally controlled sampling, while continuous sampling demands far longer records for the same accuracy. These findings provide a practical framework for optimizing data collection in wind resource assessments, balancing accuracy, temporal coverage, and resource constraints.

## 1 Introduction

Wind energy production critically depends on strengths and persistence of winds in the lower earth's atmosphere. Precise and cost-effective assessment of wind speed is crucial for evaluating wind energy potential and designing wind farms and power generators, because accurate assessments ensure that the selected site has adequate wind conditions, making the investment economically viable and optimizing energy production efficiency (Wang et al., 2022).

Quantifying wind speed characteristics, a crucial component of wind speed assessment, typically relies on analysing wind speed distribution from collected data. Ideally, long-term meteorological measurements at the proposed wind turbine locations are preferred, as they account for a broader range of wind variability. Wind speed measurements spanning four years are

typically considered suitable for short-term analysis, while datasets extending beyond this period fall into the category of long-term analysis. A ten-year dataset is generally recommended for the most accurate wind resource assessment, if available (Murthy and Rahi, 2017). However, collecting such long-term measurements is often impractical due to the time financial constraints involved, particularly in the early planning stages of wind farm development (Wais, 2016).

As a more practical alternative, wind energy potential is often assessed using wind speed data spanning a single year or a few years (Ouarda et al., 2015). A review of 46 studies revealed that 31 of them (67.4%) used wind speed time series of six years or less. However, such datasets lack year-to-year (interannual) variability, which can significantly affect wind speed and, consequently, wind power output (Jung and Schindler, 2018). For example, decadal changes in wind speed can result in a $17 \pm 2\%$ variation in potential wind energy (Zeng et al., 2019). Since wind farms typically operate for 20 to 30 years (Pryor et

al., 2020), relying on such short-term datasets without accounting for interannual variability can introduce significant biases in wind energy assessments. Additionally, short-term datasets may lack seasonal or diurnal characteristics due to sampling frequency or other factors that lead to data gaps. For instance, the Sentinel-1 Ocean wind product, aligning well with in-situ observations and reanalysis products (Khachatrian et al., 2024), revisits the same location only once every one or two days, making it unable to capture the diurnal characteristics of wind speed.


This discussion highlights a critical research gap: the optimal duration of wind observation time series required to adequately account for wind variability in resource assessments remains poorly quantified. Specifically, is one year of data, as recommended by IEC (International Electrotechnical Commission, 2019), sufficient to provide accurate assessments of wind distributions given the interannual variability of wind? Furthermore, considering the challenges in obtaining long-term

observations, if we must reply on short-term datasets that may lack interannual, seasonal, or diurnal variability, how do errors vary with the length of data time series?

This research gap has been highlighted in previous studies. For instance, Barthelmie and Pryor, (2003) and Pryor et al., (2004) evaluated the accuracy of satellite sampling in representing offshore wind speed distributions. They quantified the numbers of

satellite observations required to estimate key probability distribution parameters with an uncertainty of ±10%. Specifically, mean and Weibull scale parameter required about 60-70 random selected half-hourly observations, respectively. In contrast, the variance requires 150 observations, and the Weibull shape parameter and energy density require nearly 2000 observations, while skewness and kurtosis required 9712 and more than 10000 observations. However, these results are specific to satellite observations and may not directly apply to in-situ measurements without further analysis. In-situ measurements, such as

meteorological weather stations, are more widely distributed, accessible, and frequently used in wind energy studies (Ouarda et al., 2015; Wang et al., 2016). To the authors' knowledge, relatively few studies have examined in-situ observations, particularly those from weather stations certified by the World Meteorological Organization (WMO).

Our study aims to evaluate the potential biases and uncertainties that may arise when short-term wind speed data from WMO weather stations are used for wind energy assessments. Previous work by Barthelmie and Pryor (2003) proposed a random sampling approach to examine how sampling protocols affects the estimation of wind speed distribution parameters. However, random sampling may overlook the diurnal and seasonal cycles that are intrinsic to in-situ terrestrial wind observations and critical for reliable wind energy analysis. To address this limitation, we first compare random sampling with sampling strategies that explicitly retain diurnal and seasonal cycles. This comparison allows us to isolate and quantify the influence of temporal structures on wind speed statistics. In addition, we evaluate the practical relevance of random sampling by contrasting it with continuous sampling, that preserves the chronological sequence of wind speed data and more closely reflects real-world wind resource assessment practices. Continuous datasets, such as those from anemometer towers, are commonly used in the wind energy industry, typically covering at least one year of measurements to characterize site-specific wind conditions prior to turbine installation (Yang et al., 2024; Liu et al., 2023). By integrating these multiple sampling strategies, our study provides a comprehensive assessment of how sampling choices affect the robustness of wind energy evaluations based on limited-duration datasets.

We further investigate how results derived from reanalysis products differ from those obtained using WMO weather station data under various sampling strategies. Reanalysis products have emerged as a primary alternative for wind resource assessment, especially given the spatial and temporal limitations of traditional in-situ observations (Gil et al., 2021; Gualtieri, 2021). These datasets provide spatially continuous and temporally consistent wind speed data by assimilating observational data from multiple sources, including satellite instruments, surface synoptic observations, ships, and drifting buoys, into numerical weather prediction models (Hersbach et al., 2022). ERA5 stands out as the most widely used and up-to-date reanalysis product. We used ERA5 in our study because its strong agreement with observed wind data at turbine-relevant heights, especially across Europe and North America (Ramon et al., 2019). ERA5 provides wind speed data at both 10 m and 100 m, enabling direct analysis at typical hub heights and thus avoiding the need for extrapolation methods, such as wind profile log or power-law methods, to estimate wind speeds at hub height (e.g., Soares et al., 2020; Jung and Schindler, 2019).

The main objectives of our study are as follows:

1. To evaluate how the wind speed statistics (e.g., distribution parameters) derived from short-term WMO station data different those obtained from longer-term records.

2. To determine the optimal time series length required for accurate estimation of wind speed distribution parameters, with quantified uncertainty margins.

3. To explore whether ERA5 reanalysis products, at both 10-meter and 100-meter heights, yield consistent results with ground-based observations.

Through these objectives, we aim to enhance the understanding of the limitations and capabilities of short-term meteorological data in wind speed assessment, contributing to more reliable wind energy evaluations.

## 2 Data and Methods

### 2.1 Sampling methods

**2.1.1 Random sampling**

To determine the optimal length of wind speed series for accurately representing wind speed distribution parameters, we adopted the random sampling method proposed by Barthelmie and Pryor (2003). In our study, this approach involves comparing the distribution parameters derived from the full 16-year hourly wind speed series (referred to as the study datasets) with those obtained from randomly sampled subsets of varying lengths. Specifically, we constructed sample datasets ranging

from 720 hours (30 days) to 52,560 hours (6 years), with increments of 240 hours (10 days) increments. For each sample size, 1,000 synthetic datasets were generated by randomly selecting hourly observations with replacement from the full series using NumPy's '*random*' package.

For each generated dataset, we calculate seven parameters four common statistical descriptors (mean, standard deviation,

skewness, kurtosis), two Weibull parameters (shape and scale), and the Weibull wind power density. To evaluate the representativeness of these sampled subsets, we computed the percent error between each parameter estimated from the sample and the corresponding parameter from the full 16-year series. Specifically, we focused on the upper and lower bounds of the 90% confidence interval for each parameter across 1000 realizations at each sample size. The percent errors ($Y$) in these bounds were then modelled as a function of sample size ($n$) using non-linear least squares fitting, resulting in equations that describe

how sampling uncertainty decreases with increasing sample length ($Y = \pm exp[a \ln(n) + b]$). These fitted curves enable estimation of the minimum dataset length needed to achieve predefined error margins.

We selected 720 hours as the starting point based on its frequent use in previous wind studies (e.g., Jung and Schindler, 2019; Ouarda and Charron, 2018), while the upper limit of 52,560 hours (six years) was based on prior findings (Barthelmie and

Pryor, 2004) showing that percent errors generally stabilize before this duration.

### 2.1.2 Diurnal- and seasonality-retained sampling

We implemented two structured sampling methods to retain key temporal patterns in the wind speed data: diurnal-retained sampling and seasonality-retained sampling. In the diurnal-retained approach, each synthetic dataset consists of observations evenly distributed across four 6-hour time intervals (00:00–05:00, 06:00–11:00, 12:00–17:00, and 18:00–23:00), to preserve

daily variability. For example, when the sample size is 720, we select 180 observations from each time interval. In the seasonality-retained sampling, each dataset includes an equal number of observations from all 12 months, thereby maintaining seasonal structure. For a sample size of 720, this results in 60 observations per month. For both methods, sampling was performed with replacement, meaning the same observation could be selected in multiple realizations.

### 2.1.3 Continuous sampling

The continuous sampling method is designed to simulate real-world scenarios in which wind speed data are used in their natural temporal sequence. Unlike the random and stratified (diurnal- or seasonality-retained) sampling approaches, this method preserves the chronological order of observations by extracting time-contiguous subsets directly from the full series. Prior to sampling, linear interpolation was applied to fill any missing values. In this study, we investigated sample sizes ranging from 720 hours (approximately one month) to 103,680 hours (12 years), increasing in one-month (720-hour) increments. As

this method requires each extracted subset to be continuous, the source dataset must be longer than or equal to the target sample size. For example, given a 46-year hourly wind speed dataset, we can extract all possible one-year-long continuous sequences (i.e., using a moving window of one year), resulting in 395,089 potential samples of 8,640 hourly observations each. Due to computational constraints, we randomly selected 1,000 sequences for each sample size, in line with the approach used for the other sampling methods. The same parameter estimation procedure was then applied to these sequences to assess variability

and estimate confidence intervals.

### 2.2 Probability density distributions

In this study, we exclusively employed the two-parameter Weibull probability density function to fit wind speed data. The function is expressed as Eq. (1):

$$p(v) = \left(\frac{k}{c}\right)\left(\frac{v}{c}\right)^{k-1} e^{-\left(\frac{v}{c}\right)^k} , \tag{1}$$

where $v$ represents the wind speed, $k$ is the shape parameter, and $c$ is the scale parameter. The Weibull distribution is characterized by two key parameters: the dimensionless shape parameter, which determines the curve's shape, and the scale parameter, which adjusts the distribution along the wind speed axis. The distributions vary with different values of $k$ and $c$, making it a popular choice for approximating observed wind speed frequencies (Wais, 2017; Ouarda and Charron, 2018; Carta et al., 2009).


To estimate the Weibull parameters, we used the '*weibull_min.fit*' function from Python's '*scipy.stats*', employing the maximum likelihood estimation (MLE) method. MLE is preferred for its superior performances in parameter selection (Mohammadi et al., 2016). This '*weibull_min.fit*' function is particularly useful for iterative experiments requiring repeated Weibull distribution fitting, such as those with thousands of iterations.


We focused on the first four moments of the distributions: mean, standard deviation, skewness, kurtosis, and the Weibull shape and scale parameters, chosen for their importance in wind resource assessment. The standard deviation indicates wind speed variability, while skewness and kurtosis provide insights into asymmetry and extreme values in the distribution. We calculated the mean and standard deviation using Python's 'NumPy' package, and the other parameters with '*scipy.stats*'.

## 2.3 Wind resource assessment method

We used the Weibull wind power density to represent wind resources at a specific location. The Weibull wind power density is calculated using the estimated Weibull $k$ and $c$ parameters, and is given by the Eq. (2):

$$E = \frac{1}{2}\rho c^3 \Gamma\left(1 + \frac{3}{k}\right), \qquad (2)$$

where $E$ represents the wind power density (W m$^{-2}$), $\rho$ is air density (with 1.225 kg m$^{-3}$, the standard air density provided by IEC, used for calculation), and $\Gamma$ is the gamma function.

We chose the Weibull wind power density in our study for two main reasons. First, wind power density measures the amount of kinetic energy in airflow passing through a unit area, which can be converted into wind energy. It is a critical metric for evaluating wind resources and has been widely adopted in previous studies (e.g., Wang et al., 2022; Mohammadi et al., 2016). Second, the Weibull wind power density can be easily derived from the scale and shape parameters of the Weibull distribution, simplifying the calculation process.

Given that our objective is to determine which dataset—specifically, which time series length—most accurately represents long-term wind conditions, the use of Weibull wind power density enables us to compare how the shape and scale parameters vary with datasets of different lengths. This approach helps us more effectively identify the time series that best captures long-term wind resource variability.

## 2.4 Data sources

### 2.4.1 In-situ observations from weather stations

In this study, we first utilized weather station observations from the Norwegian Meteorological Institute (MET Norway). This data, accessed via their API (https://frost.met.no/observations/v0.jsonld?; last accessed 8 February 2025), offers hourly wind speed resolution over long periods, suitable for analysing interannual variability, as wind assessments typically need at least hourly resolution (Jung and Schindler, 2019).

We aimed to compare wind distribution parameters from short-term data with long-term series that include interannual variability. We prioritized weather stations with the longest hourly data series, retaining years with at least 8,600 hourly observations (97.9% of the possible 8,760 or 8,784 hours annually).

We identified five stations with over 16 years of hourly data: SN50500 (18 years), SN44080 (16 years), SN42160 (20 years), SN38140 (24 years), and SN35860 (17 years). Details are in Table 1, and their locations in southern Norway are shown in Fig. 1. We standardized the data to 16 years per station, omitting years with fewer observations for consistency.

Using the same years across all stations was not feasible due to data availability differences, so the years analysed varied. Table S1 details the selected years and percentage of hourly observations. The year with the fewest observations had 8,648 hours (98.45% coverage), and the average yearly count was 8,744 hours (99.54% coverage).


Additionally, to complement the main analysis conducted on above five Norwegian stations, we used two additional stations located in Copenhagen Airport (Denmark) and Leuchars (Scotland, UK) from another dataset, HadISD, version v3.4.2.202501p (https://www.metoffice.gov.uk/hadobs/hadisd/; last accessed 14 June 2025; Dunn et al. 2016). Both sites provide 46 years (1979-2024) of hourly wind speed observations with an average data coverage of 99.2% annually (minimum

yearly data coverage is 95.7% due to untimely updated data for 2024). The data coverage of each year is shown in Fig. S1.

**Table 1: Details of weather stations used in this study.**

| Station ID | Location | Data source | WMO number | Latitude | Latitude of ERA5 grid | Longitude | Longitude of ERA5 grid | Height above mean sea level | Elevation of ERA5 grid |
|---|---|---|---|---|---|---|---|---|---|
| SN50500 | Flesland | | 1311 | 60.2892º N | 60.25º | 5.2265º E | 5.25º | 48 m | 0.3 m |
| SN44080 | Obrestad Fyr | | 1412 | 58.6592º N | 58.75º | 5.5553º E | 5.50º | 24 m | 5.6 m |
| SN42160 | Lista Fyr | MET Norway | 1427 | 58.1090º N | 58.00º | 6.5675º E | 6.50º | 14 m | 127.1 m |
| SN38140 | Landvik | | 1464 | 58.3400º N | 58.25º | 8.5225º E | 8.50º | 6 m | 55.4 m |
| SN35860 | Lyngør Fyr | | 1467 | 58.6362º N | 58.75º | 9.1478º E | 9.25º | 4 m | 43.9 m |
| 061800-99999 | Kastrup | HadISD | / | 55.618º N | / | 12.656º E | / | 5.2 m | / |
| 031710-99999 | Leuchars | | / | 56.373º N | / | -2.868º E | / | 11.6 m | / |

Note: As the last two stations (Kastrup and Leuchars) were added specifically for the sensitivity analysis discussed in Section 4.1, they were excluded from the comparison with ERA5.

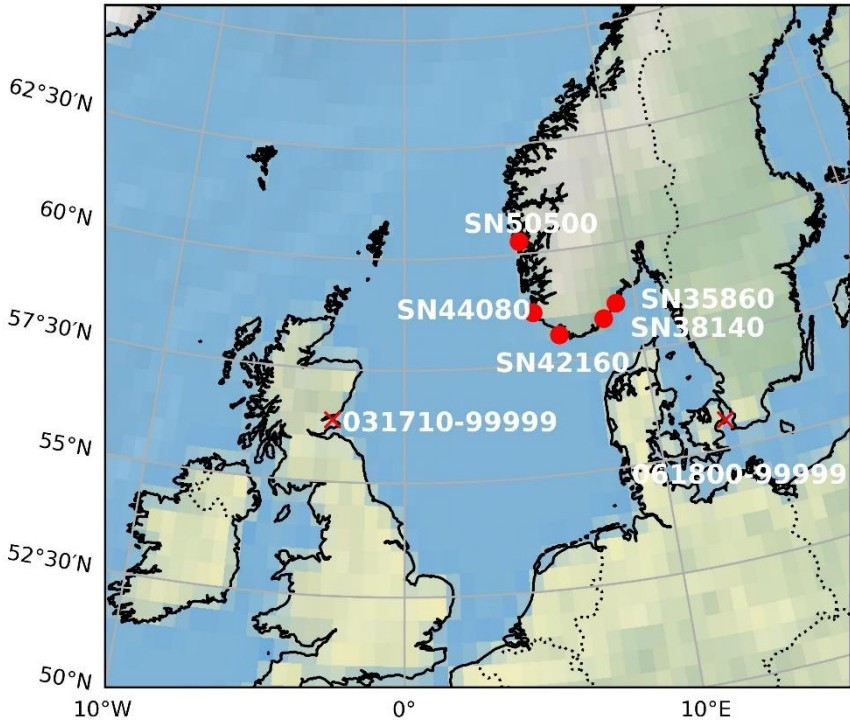


**Figure 1: Distribution of the weather stations used in this study.**

### 2.4.2 ERA5 reanalysis

For the ERA5 reanalysis products, we downloaded the "10m u-component of wind," "10m v-component of wind," "100m u-
component of wind," and "100m v-component of wind" variables from the Copernicus Climate Data Store
(https://cds.climate.copernicus.eu/datasets/reanalysis-era5-single-levels?tab=download; last accessed 8 February 2025). We
calculated the wind speed at 10 m and 100 m by taking the square root of the sum of the squares of the u-component and v-
component of wind. We used the ERA5 grid point closest to the location of each station, as indicated in Table 1.

### 3 Results

**3.1 Can random sampling replace diurnal cycle-retained or seasonality-retained sampling?**

The five Norwegian stations exhibit distinct diurnal and seasonal variations (Fig. S1-S2). To assess whether random sampling
can serve as a substitute for diurnal cycle-retained or seasonality-retained sampling, we compared the 90% confidence intervals
(CIs) of distribution parameters derived from each method, rather than on single-point parameter estimates. This comparison
can also help understand how sampling strategy affects uncertainty.

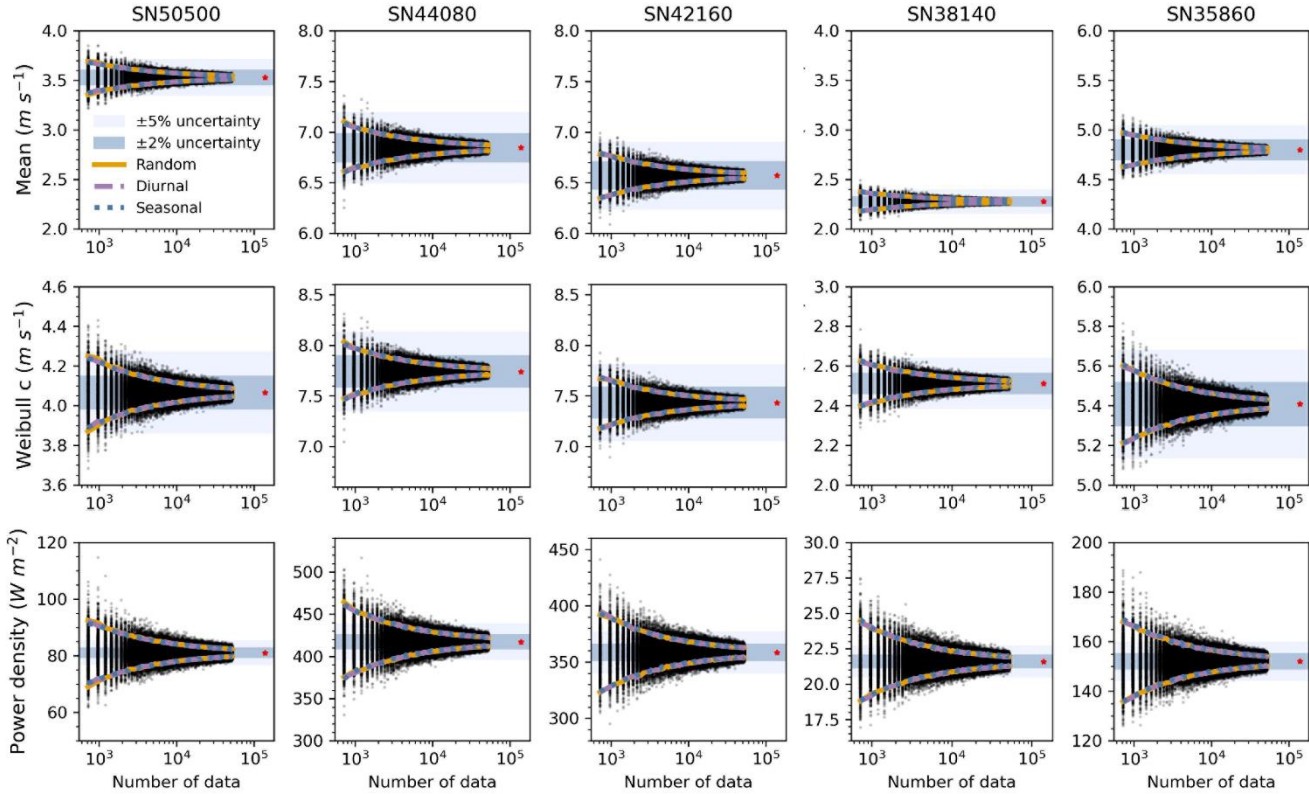

**Figure 2: Estimates of mean wind speed, Weibull scale parameter, and power density from three sampling strategies, based on in-situ observations from five Norwegian stations.** The 90% confidence intervals (CIs) are shown for each sampling method: random (orange), diurnal-cycle-retained (purple dashed), and seasonality-retained (blue dotted). Each black dot represents a parameter estimate from a single sampling realization of random sampling; corresponding realizations for the other two methods are not shown. Sample sizes range from 720 to 52,560 (30 days to 6 years), increasing in 240-hour (10-day) increments, with 1,000 realizations per size. Red asterisks indicate the reference values from the full 16-year hourly dataset (see Table 2). Shaded areas represent ±2% (dark blue) and ±5% (light blue) deviation ranges from full-series values.


To visually compare the uncertainty ranges between the sampling methods, Fig. 2 and Fig. S4 presents the 90% confidence intervals (CIs) derived from each approach. It is evident that the intervals from random sampling largely overlap with those from diurnal and seasonality-retained sampling. To quantify these differences, we calculated the CI differences (Fig. S5) and the root mean square error (RMSE) of these differences (Table S2). Most parameter differences fluctuate around zero, with magnitudes generally within ±0.2; power density is the only parameter showing larger fluctuations, within ±3. These

differences tend to decrease as sample density increases (Fig. S5). Power density also exhibits the largest RMSE, likely due to its broader value range (from tens to hundreds), while the shape parameter show the smallest RMSE (Table S2).

We further examined whether similar results hold for ERA5 wind speed data at 100 meters, which better reflect turbine-relevant altitudes and help address the scarcity of high-elevation measurements. Similar CI overlaps were observed (Fig. 3, S6). The mean RMSEs of the differences of parameters from the ERA5 100-meter (0.4896 for diurnal-retained and 1.1010 seasonal-retained) were comparable to those from in-situ : 0.2865 (diurnal-retained) and 0.3903 (seasonality-retained). The higher values were primarily driven by power density differences (Table S2). A similar pattern in the 90% confidence interval differences among the three sampling strategies is observed in the ERA5 100 m dataset and the in-situ observations (Fig. S7). Based on these findings, we conclude that random sampling is a viable method for estimating wind distribution parameters, both at surface and turbine hub heights. Therefore, we adopted random sampling in subsequent analyses to determine the optimal sample size for capturing long-term wind characteristics.

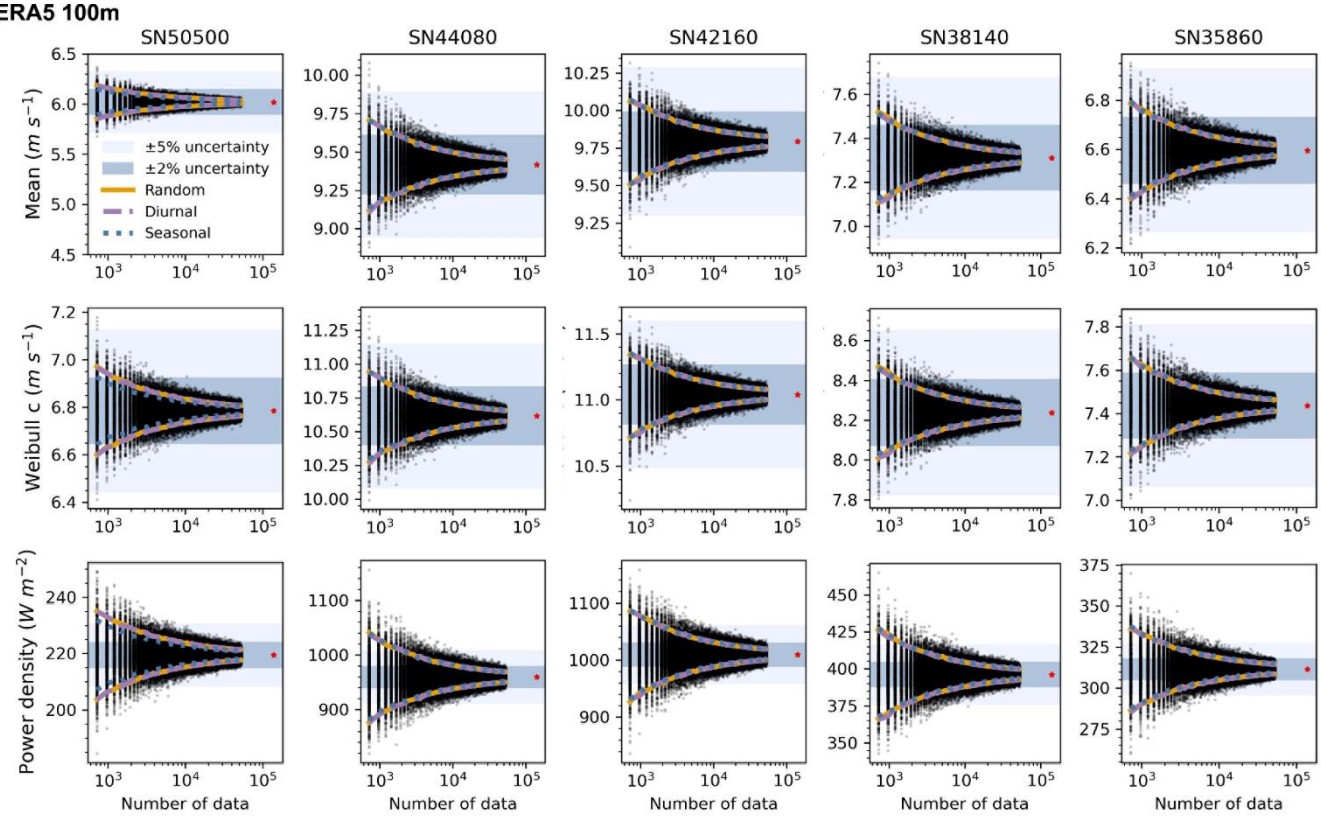

**Figure 3: Estimates of mean wind speed, Weibull scale parameter, and power density from three sampling strategies, based on ERA5 100-meter data.** Sampling methods and visualization are consistent with Figure 2. Red asterisks indicate values from the full 16-year ERA5 100 m dataset. Shaded areas represent ±2% (dark blue) and ±5% (light blue) deviation ranges from full-series values.

### 3.2 Effects of sample size on estimating wind distribution parameters

We investigated how sample size affects the accuracy of wind distribution parameters. Despite differences in wind conditions (Table 2; Fig. S8), all five Norwegian stations exhibited consistent patterns.

**Table 2: Distribution parameters and Weibull power density of five Norwegian stations, derived from the entire datasets.** Note: for ERA5 products, the station ID indicates the corresponding grid point location.

| Data products | Station ID | Mean (m s$^{-1}$) | Std. dev. (m s$^{-1}$) | Skewness | Kurtosis | Shape $k$ | Scale $c$ (m s$^{-1}$) | Power density (W m$^{-2}$) |
|---|---|---|---|---|---|---|---|---|
| In-situ weather stations | SN50500 | 3.53 | 2.66 | 1.12 | 1.81 | 1.51 | 4.07 | 81.08 |
| | SN44080 | 6.85 | 3.94 | 0.76 | 0.45 | 1.83 | 7.74 | 417.34 |
| | SN42160 | 6.57 | 3.68 | 0.65 | 0.34 | 1.88 | 7.43 | 358.49 |
| | SN38140 | 2.28 | 1.61 | 0.92 | 1.28 | 1.42 | 2.51 | 21.61 |
| | SN35860 | 4.80 | 2.88 | 0.79 | 0.47 | 1.74 | 5.41 | 152.15 |
| ERA5 (10 meter) | SN50500 | 4.82 | 2.45 | 0.30 | -0.68 | 2.07 | 5.44 | 126.73 |
| | SN44080 | 7.58 | 3.74 | 0.35 | -0.36 | 2.13 | 8.55 | 478.87 |
| | SN42160 | 8.04 | 3.74 | 0.32 | -0.28 | 2.28 | 9.07 | 539.59 |
| | SN38140 | 4.74 | 2.27 | 0.45 | -0.15 | 2.20 | 5.35 | 113.61 |
| | SN35860 | 4.50 | 2.19 | 0.48 | -0.06 | 2.16 | 5.08 | 98.77 |
| ERA5 (100 meter) | SN50500 | 6.02 | 2.71 | 0.22 | -0.48 | 2.36 | 6.78 | 219.44 |
| | SN44080 | 9.42 | 4.83 | 0.40 | -0.29 | 2.03 | 10.61 | 959.38 |
| | SN42160 | 9.79 | 4.72 | 0.35 | -0.18 | 2.18 | 11.04 | 1009.61 |
| | SN38140 | 7.31 | 3.31 | 0.31 | -0.07 | 2.33 | 8.24 | 396.08 |
| | SN35860 | 6.60 | 3.21 | 0.37 | -0.13 | 2.15 | 7.44 | 311.57 |

We found that, as sample size increased, the 90% confidence intervals (CIs) for all parameters narrowed, though the rate of convergence varied. The mean, standard deviation, and Weibull $k$ and $c$ parameters stabilized quickly, within ±5% error margins even at 720 hourly observations (Fig. 2, S4). In contrast, power density showed greater variability, and skewness and kurtosis were far less robust, remaining beyond ±5% even after six years of hourly data due to their sensitivity to distribution tails and extremes.

To assess systematic bias, we examined the median values across 1,000 resampling iterations(Fig. S9). Skewness and especially kurtosis showed notable underestimation at low sample sizes. At 720 observations, median skewness was over 2% lower, and kurtosis more than 25% lower than the full-series baseline. The kurtosis bias remained above 10% until sample size exceeded 2,160 hours, and SN50500 required 22,080 observations (~2.5 yrs) to reduce error to within 10%. In contrast, other parameters varied by less than 1% across all sample sizes.

### 3.3 Determine an effective sample size for capturing overall wind characteristics

To determine the optimal sample size for capturing wind characteristics, we analysed the relationship between percent errors and sample sizes (Fig. 4-5). Percent error measures discrepancies between parameters from the full dataset and smaller subsets. Based on the 90% CIs derived from 1,000 realizations of random sampling of in-situ observations (orange lines in Fig. 2 &

Fig. S4), we computed percent errors of CI bounds and fitted power-law equations to describe their dependence on sample size. These fitted equations are summarized in Table 3 and allow extrapolation of error margins for any given sample size.

As expected, percent error decreases with increasing sample size, though the rate and extent vary across parameters. For most stations, 720 hourly observations are sufficient to constrain the percent errors within ±7% for the mean, standard deviation, and Weibull parameters (Fig. 4). In contrast, higher-order statistical moments such as skewness and kurtosis, as well as power density, show much larger errors under the same sampling conditions, with deviations ranging from ±10% up to ±150%, depending on the station. These parameters show greater variability across stations, with error differences of 4.6% for power density, 18.1% for skewness, and 154.2% for kurtosis, compared to less than 1.5% for others. Errors decrease quickly below 400 observations and more slowly above (Fig. 5). About 200 observations can achieve ±10% error for the mean, standard deviation, and Weibull parameters (Fig. 5). To facilitate practical use, we calculated the minimum sample sizes required to achieve ±10%, ±5%, ±2%, and ±1% error margins for each parameter at each station (Table 4). For example, ±5% accuracy requires 459 observations for the mean, 470 for the Weibull scale (20 days), 796 for standard deviation (34 days), and 4,031 for power density. Achieving ±2% and ±1% error requires 6-fold and 24-fold of observations than ±5%, respectively. Skewness and kurtosis are especially data-intensive due to their sensitivity to distribution tails. For instance, SN38140 needs 177,390 observations (20 years) for ±10% error, while SN50500 needs 1,541,437 observations (176 years).

We also observe regional differences in sample requirements. Stations with higher wind speed variability, but lower skewness and kurtosis tend to require fewer samples. For example, SN50500 and SN38140, with the highest skewness and kurtosis, require more observations. Power density has the largest regional difference (max/min ratio = 2.1), while the Weibull shape shows the least (1.2). Skewness and kurtosis are sensitive to wind characteristics, with required samples increasing 3.96-6.1 and 8.69-13.16, respectively, when error margins decrease from ±10% to ±1%.

**Table 3. Fitted equations describing the relationship between the percent error ($Y$) and sample size ($n$), based on random sampling results from five in-situ weather stations.** Each equation corresponds to a power-law fit of the 90% confidence interval (CI) bounds, positive (P) and negative (N), for each parameter, across sample sizes from 720 to 52,560 hours.

| Parameters | SN50500 | SN44080 | SN42160 | SN38140 | SN35860 |
|---|---|---|---|---|---|
| Mean (P) | Y=exp[-0.507ln(n)+4.888] | Y=exp[-0.503ln(n)+4.579] | Y=exp[-0.497ln(n)+4.497] | Y=exp[-0.496ln(n)+4.724] | Y=exp[-0.494ln(n)+4.536] |
| Mean (N) | Y=-exp[-0.511ln(n)+4.929] | Y=-exp[-0.494ln(n)+4.491] | Y=-exp[-0.498ln(n)+4.504] | Y=-exp[-0.500ln(n)+4.758] | Y=-exp[-0.501ln(n)+4.601] |
| Std. dev (P) | Y=exp[-0.497ln(n)+5.045] | Y=exp[-0.503ln(n)+4.579] | Y=exp[-0.486ln(n)+4.692] | Y=exp[-0.497ln(n)+4.971] | Y=exp[-0.489ln(n)+4.748] |
| Std. dev (N) | Y=-exp[-0.509ln(n)+5.169] | Y=-exp[-0.494ln(n)+4.491] | Y=-exp[-0.500ln(n)+4.838] | Y=-exp[-0.503ln(n)+5.033] | Y=-exp[-0.504ln(n)+4.904] |
| Skewness (P) | Y=exp[-0.452ln(n)+6.610] | Y=exp[-0.495ln(n)+6.434] | Y=exp[-0.483ln(n)+6.523] | Y=exp[-0.482ln(n)+6.579] | Y=exp[-0.488ln(n)+6.254] |
| Skewness (N) | Y=-exp[-0.471ln(n)+6.807] | Y=-exp[-0.502ln(n)+6.522] | Y=-exp[-0.496ln(n)+6.665] | Y=-exp[-0.506ln(n)+6.825] | Y=-exp[-0.509ln(n)+6.475] |
| Kurtosis (P) | Y=exp[-0.436ln(n)+8.521] | Y=exp[-0.493ln(n)+8.449] | Y=exp[-0.474ln(n)+8.746] | Y=exp[-0.469ln(n)+7.971] | Y=exp[-0.488ln(n)+8.273] |
| Kurtosis (N) | Y=-exp[-0.451ln(n)+8.673] | Y=-exp[-0.500ln(n)+8.540] | Y=-exp[-0.485ln(n)+8.869] | Y=-exp[-0.496ln(n)+8.254] | Y=-exp[-0.507ln(n)+8.472] |
| Weibull $k$ (P) | Y=exp[-0.508ln(n)+4.902] | Y=exp[-0.503ln(n)+4.845] | Y=exp[-0.503ln(n)+4.907] | Y=exp[-0.509ln(n)+4.994] | Y=exp[-0.51ln(n)+4.919] |
| Weibull $k$ (N) | Y=-exp[-0.491ln(n)+4.721] | Y=-exp[-0.493ln(n)+4.731] | Y=-exp[-0.484ln(n)+4.696] | Y=-exp[-0.501ln(n)+4.906] | Y=-exp[-0.493ln(n)+4.735] |
| Weibull $c$ (P) | Y=exp[-0.507ln(n)+4.864] | Y=exp[-0.503ln(n)+4.580] | Y=exp[-0.496ln(n)+4.494] | Y=exp[-0.497ln(n)+4.782] | Y=exp[-0.494ln(n)+4.55] |
| Weibull $c$ (N) | Y=-exp[-0.512ln(n)+4.906] | Y=-exp[-0.495ln(n)+4.505] | Y=-exp[-0.498ln(n)+4.506] | Y=-exp[-0.501ln(n)+4.824] | Y=-exp[-0.501ln(n)+4.619] |
| Power density (P) | Y=exp[-0.508ln(n)+6.011] | Y=exp[-0.505ln(n)+5.689] | Y=exp[-0.495ln(n)+5.547] | Y=exp[-0.500ln(n)+5.854] | Y=exp[-0.493ln(n)+5.614] |
| Power density (N) | Y=-exp[-0.509ln(n)+6.014] | Y=-exp[-0.492ln(n)+5.560] | Y=-exp[-0.497ln(n)+5.566] | Y=-exp[-0.497ln(n)+5.813] | Y=-exp[-0.5ln(n)+5.674] |

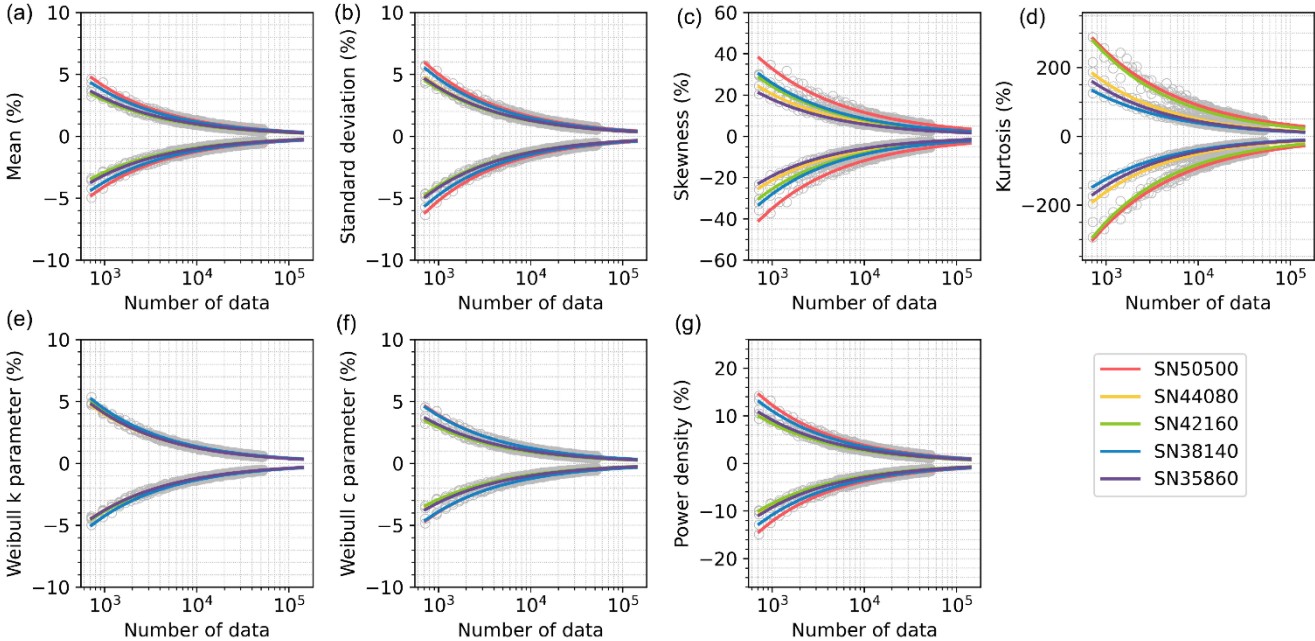

**Figure 4: The relationship between the percent error (Y) and sample size (n) based on hourly observations ranging from $n$ = 720 (30 days) to $n$ = 140,160 (16 years) across five stations.** The equations of fits here are shown in Table 3. Grey circles indicate the values used to fit the 90% confidence intervals for the percent error shown.

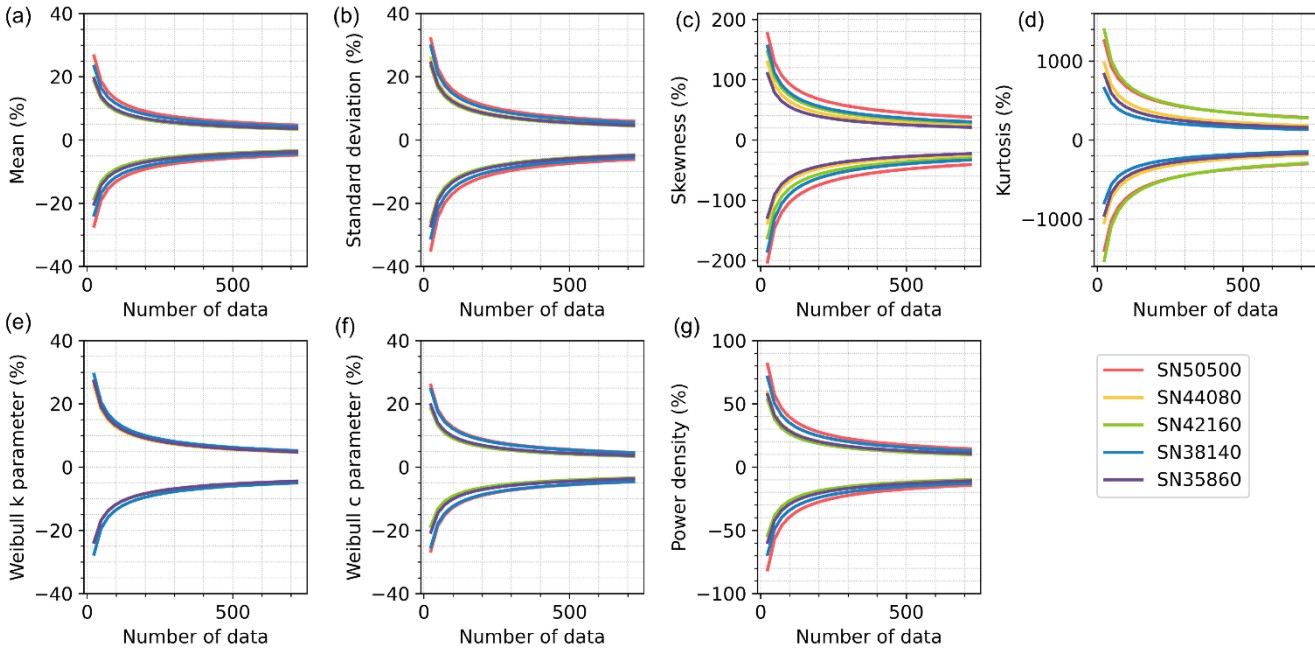

310

**Figure 5: Same as Fig. 4, but the hourly observations ranging from $n$= 24 (1 day) to $n$= 720 (30 days) across five stations.** These intervals are calculated using the same fits as shown in Fig. 4.

**Table 4. Required number of randomly selected in-situ observations (unit: hours) to obtain an estimate within ±10%, ±5%, ±2%, and ±1% of the parameters from the entire observed time series (157,465 data points), calculated at the 90% confidence level.** The fits to get the required data density are shown in Table S2.

| Error margins | Location | Mean | Std. dev. | Skewness | Kurtosis | Shape $k$ | Scale $c$ | Power density |
|---|---|---|---|---|---|---|---|---|
| ±10% | SN50500 | 170 | 279 | 14297 | 1541437 | 166 | 162 | 1489 |
| | SN44080 | 92 | 162 | 4505 | 262169 | 157 | 93 | 813 |
| | SN42160 | 83 | 160 | 6658 | 801270 | 177 | 84 | 709 |
| | SN38140 | 135 | 228 | 7673 | 177390 | 198 | 153 | 1211 |
| | SN35860 | 98 | 175 | 3611 | 204844 | 169 | 101 | 853 |
| | average | 116 | 201 | 7349 | 597422 | 174 | 119 | 1015 |
| ±5% | SN50500 | 659 | 1087 | 63795 | 7545102 | 649 | 629 | 5836 |
| | SN44080 | 365 | 655 | 17944 | 1058755 | 623 | 368 | 3202 |
| | SN42160 | 335 | 640 | 26968 | 3458621 | 700 | 338 | 2859 |
| | SN38140 | 541 | 905 | 30229 | 777573 | 774 | 610 | 4840 |
| | SN35860 | 393 | 691 | 14084 | 847284 | 657 | 404 | 3417 |
| | average | 459 | 796 | 30604 | 2737467 | 681 | 470 | 4031 |
| ±2% | SN50500 | 3956 | 6576 | 484327 | 61581562 | 3936 | 3770 | 35501 |
| | SN44080 | 2256 | 4165 | 111517 | 6790761 | 3853 | 2276 | 19931 |
| | SN42160 | 2113 | 4008 | 174520 | 23905124 | 4321 | 2131 | 18057 |
| | SN38140 | 3379 | 5593 | 200542 | 5484926 | 4689 | 3793 | 30218 |
| | SN35860 | 2445 | 4262 | 88940 | 5535245 | 3956 | 2513 | 21623 |
| | average | 2830 | 4921 | 211970 | 20659524 | 4151 | 2897 | 25066 |
| ±1% | SN50500 | 15531 | 25766 | 2244402 | 301432368 | 15383 | 14785 | 139117 |
| | SN44080 | 8944 | 16876 | 444166 | 27700221 | 15295 | 9032 | 81625 |
| | SN42160 | 8503 | 16046 | 733004 | 103184595 | 17126 | 8585 | 72806 |
| | SN38140 | 13574 | 22191 | 844568 | 24042683 | 18315 | 15117 | 120783 |
| | SN35860 | 9757 | 16870 | 368113 | 22895088 | 15391 | 10011 | 88205 |
| | average | 11262 | 19550 | 926851 | 95850991 | 16302 | 11506 | 100507 |

### 3.4 Does ERA5 reanalysis (10 m and 100 m) show similar results with in-situ observations?

To assess the consistency of reanalysis data with in-situ measurements, we compared ERA5 (10 m and 100 m) in-situ observations. At four out of five stations, ERA5 overestimated mean wind speeds in both the full time series (Table 2) and sampling experiments (Fig. 6 & Fig. S10), likely due to an overrepresentation of low-to-moderate wind speeds (Fig. S8). This bias also led to overestimation of the Weibull scale parameter at stations with higher wind speeds and underestimation at those with lower speeds. Additionally, the Weibull shape parameter was consistently higher in ERA5, often exceeding 2, indicating a potential bias in overestimating high wind events. These biases collectively contributed to systematic overestimation in Weibull power density (Table 2 & Fig. 6 & Fig. S10).

Both in-situ and ERA5 distributions were positively skewed (Fig. S8), but in-situ data had higher skewness (Table 2). ERA5consistently showed lower skewness (Fig. S10). For kurtosis, in-situ observations show positive values (Table 2),

indicating more peaked distributions, whereas ERA5 exhibited negative values, reflecting flatter, less variable distribution. The largest divergence was observed at SN50500 and SN38140 (Fig. S10), where in-situ kurtosis varied substantially, while ERA5 values remained comparatively uniform (Fig. S10).

These differences influenced sample size requirements. For mean, standard deviation, Weibull scale, and power density, ERA5

(10m) generally required fewer data points to achieve the same error margins thresholds (Table S3). However, for tail-sensitive parameters like shape, skewness, and kurtosis, ERA5 require larger sample size. Additionally, ERA5 results showed lower inter-station variability, as indicated by overlapping percent-error curves (Fig. S11-S12). The equations used to estimate percent errors under different sample sizes for ERA5 10 m are summarized in Table S4.

We further analysed the ERA5 100 m dataset, which aligns more closely with hub heights. As shown in figures S13-S14, most parameters had similar data density requirements to those at ERA5 10 m dataset, though it can vary by station. For instance, SN42160 had the highest error in the 10-meter dataset, while SN35860 showed nearly double the error under the same density. Table S5 summarizes the required sample sizes, showing broadly similar patterns across both heights, but the 100-meter dataset consistently required more data for the shape parameter. The equations used to estimate the required sample sizes for ERA5

100 m are summarized in Table S6.

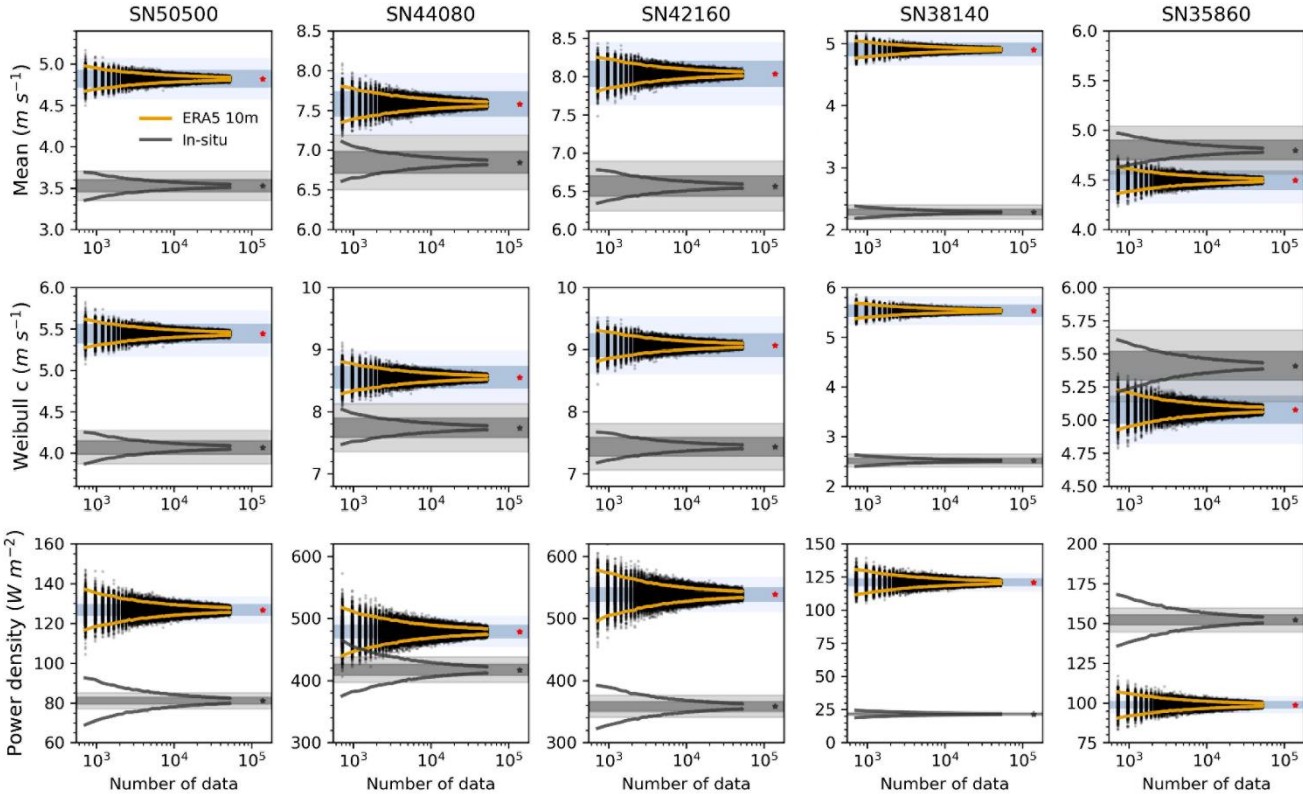

**Figure 6: Estimates of mean wind speed, Weibull scale parameter, and power density based on random sampling of ERA5 10-meter reanalysis data (black dots) across five Norwegian stations.** The sampling strategy is consistent with Figure 2. The 90% confidence intervals (CIs) are shown as orange lines (ERA5) and grey lines (in-situ observations). Red asterisks denote reference values derived from the full 16-year ERA5-10m dataset; grey asterisks represent the corresponding values from in-situ observations. Blue shading represents ±2% (dark) and ±5% (light) uncertainty margins around ERA5-10m reference values, while grey shading indicates the same margins around in-situ reference values.

## 4 Discussions and Implications

### 4.1 Sensitivity to sampling strategy and climatic non-stationarity

In wind energy assessments, continuous sampling is more commonly used than random sampling because it preserves temporal structure and seasonal variability in wind speed time series, and most importantly, only long-term data are not available. However, continuous sampling may also introduce systematic bias, particularly over short durations, due to temporal autocorrelation and underlying climatic non-stationarity. To investigate the extent of this effect and assess the generalizability of random sampling, we conducted a sensitivity analysis using 46 years (1979–2024) of hourly wind speed data from two coastal meteorological stations: Copenhagen Airport (061800-99999, Denmark) and Leuchars (031710-99999, Scotland).

These sites were chosen for their long-term records and meteorological similarity to the five Norwegian locations analysed earlier. Copenhagen station exhibits a long-term decreasing wind speed trend (Fig. S1), consistent with broader global observations (Zeng et al., 2019).

Our results show that continuous sampling generally requires significantly longer periods to achieve the same level of uncertainty in estimated distribution parameters compared to random sampling (Fig. 7). This discrepancy arises because random sampling draws from multiple years, thereby capturing a wider range of interannual variability and reducing exposure to temporal clustering. Consequently, the 90% confidence intervals (CIs) under random sampling are symmetric for all parameters, while under continuous sampling, only the CIs for mean wind speed, Weibull scale parameter, and power density are symmetric. Shape-sensitive parameters, including standard deviation, skewness, kurtosis, and especially the Weibull shape parameter, exhibit pronounced asymmetries under continuous sampling, particularly at short durations (<2 years). This suggests that the presence of systematic climatic anomalies in continuous subsets may bias shape estimation.

These findings support earlier recommendations by Murthy et al. (2017), who advocate using at least four to ten years of data for reliable wind energy assessments. Our results suggest that when using continuous sampling, at least five years of data may be required to achieve ±10% relative uncertainty in power density estimates, although this threshold is site-specific (e.g., Copenhagen station requires more than 10 years). We further recommend that random sampling be considered as a complementary tool to identify potential biases in short-term continuous assessments.

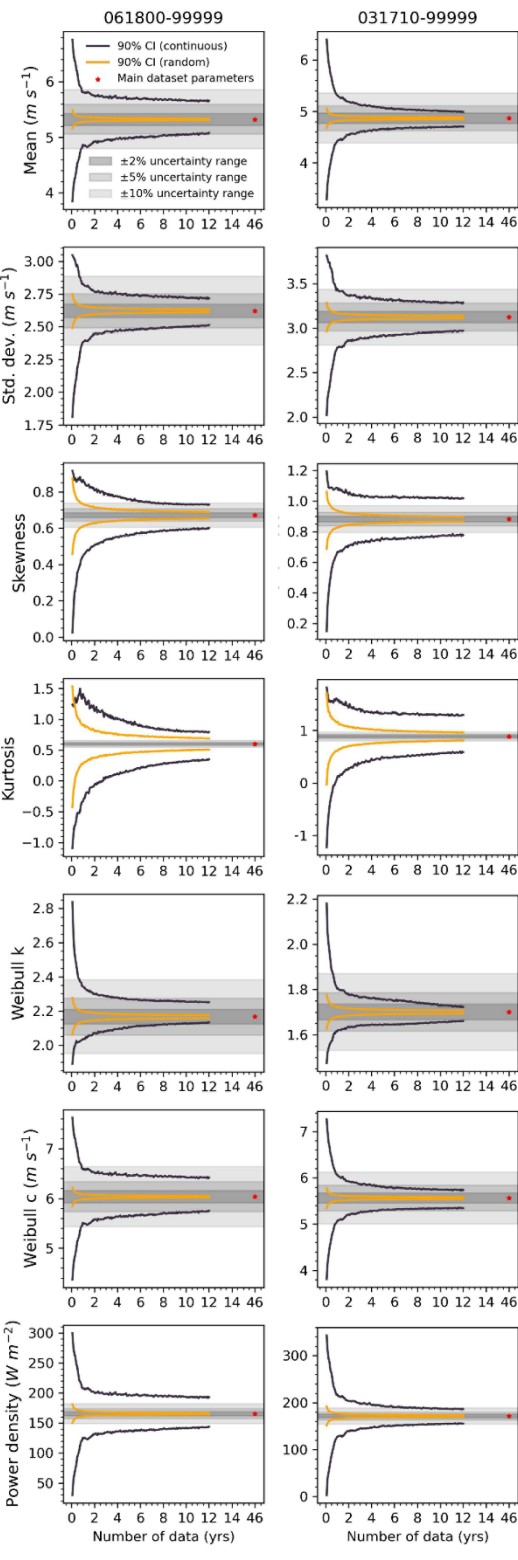

**Figure 7: distribution parameters and Weibull power density derived from random sampling (orange lines) and continuous sampling (black lines), based on in-situ measurements from weather stations.** Asterisks indicate values computed from the full 46-year dataset. Values for sample lengths between 14 and 46 years are omitted for visual clarity. Details of the experimental setup and sampling procedures are provided in the Methods section.

The uncertainty bounds acquired by the random sampling in this study provided exhibit robustness and are applicable to all remotely sensed wind speed data series (Barthelmie and Pryor, 2003). Specifically, they reached this conclusion by finding a similar required sample size with an uncertainty of ±10% from five different locations, including Denmark, eastern North Pacific, the Gulf of Mexica, the Gulf of Alaska, and the western Atlantic (Barthelmie and Pryor, 2003; Pryor et al., 2004). However, upon replicating their methods using in-situ wind speed measurements from WMO stations, we are reluctant to draw the same conclusion. Although when using the same error margin (±10%) as Barthelmie and Pryor, (2003), we obtain similar results. As the error margins narrow (from ±10% to ±1%), the discrepancy among stations becomes significant. Therefore, we suggest that the uncertainty bounds presented in Table 3 exhibit robustness and are applicable only under higher error margins, such as those exceeding ±10%. Additionally, lower moments and two Weibull parameters showed higher robustness.

Furthermore, although we provided the uncertainty bounds for datasets with fewer than 720 samples, it is important to note that we calculated these values based on an exponential function fitted to the results derived from 720 to 52,560 points. As a result, the curve may be biased due to the potential asymmetry in the distribution of the parameters (Barthelmie and Pryor, 2003).

Our results indicated that ERA5 tends to overestimate the mean and Weibull scale parameters. Discrepancies between ERA5 and observational data are unsurprising, as previous studies have noted differences in magnitude and trends (Zhou et al., 2021; Torralba et al., 2017). These discrepancies can be partly attributed to ERA5 not assimilating in-situ land observations and the inherent limitations of the ERA5 reanalysis (Hersbach et al., 2020), such as its inability to accurately reproduce mesoscale dissipation rates (Bolgiani et al., 2022). Additionally, modern data assimilation systems still struggle to adequately correct the inevitable errors in model-generated guess fields at these smaller scales (Wang and Sardeshmukh, 2021). Consequently, ERA5 may underestimate variability and fail to capture local extremes observed in in-situ data, leading to discrepancies in parameters like skewness and kurtosis. For instance, at stations SN50500 and SN38140, in-situ data show significantly more wind observations close to zero compared to ERA5 datasets, resulting in distinct wind characteristics such as differing skewness and kurtosis.

## 4.2 Evaluation of global wind atlas estimates against observations

Since the publication of the first European Wind Atlas in 1989 (Dörenkämper et al., 2020), the wind atlas methodology has been widely adopted for regional wind resource assessments, including in countries such as Finland (Tammelin et al., 2013)

and Greece (Kotroni et al., 2014). The Global Wind Atlas (GWA), developed by the Technical University of Denmark, applies the well-established numerical wind atlas method to downscale coarse-resolution reanalysis data to microscale levels. This is achieved using linearized flow models and topographic corrections based on the WAsP model. GWA provides publicly accessible estimates of mean wind speed and power density, which have been used in applications such as bias correction of reanalysis data for wind power simulations (Gruber et al., 2022).

Given the energy-focused perspective of this study, it is relevant to compare our results with GWA estimates. We extracted GWA values at the nearest grid points for selected stations and compared them with observational estimates based on the full time series. Table S7 presents this comparison, focusing on two key metrics in wind energy assessments: mean wind speed and power density. The results show that GWA consistently overestimates both wind speed and power density relative to our station-based observations.

One likely explanation for this discrepancy lies in the different ways topographic effects are incorporated. As described by Davis et al. (2023), the GWA estimates the predicted wind climate (PWC) by applying high-resolution topographic perturbations to the generalized wind climate which is based on coarse reanalysis fields. The PWC is represented by a set of Weibull distributions and directional frequencies for each of 12 directional sectors, and these are used to calculate derived variables such as mean wind speed and power density.

### 4.3 Implications

Both onshore and offshore sites exhibit seasonal variations, with onshore and near-coast locations often experiencing significant diurnal cycles (Barthelmie and Pryor, 2003; Barthelmie et al., 1996; Ashkenazy and Yizhaq, 2023). Our findings indicate that random sampling can effectively analyse wind distribution parameters, even when dealing with discontinuous data that lacks explicit diurnal or seasonal cycle information. This is particularly important given the challenges associated with accurately collecting data that reflects these cycles; factors such as anemometer malfunctions, site relocations, and other disruptions can create gaps in the wind speed data series, leading to non-continuous records (Liu et al., 2024). For instance, the Sentinel-1 Level 2 OCN ocean wind field product (1 km resolution), while performing well in offshore areas, has a revisit frequency of one to two days that may not sufficiently capture rapid temporal variations (Khachatrian et al., 2024).

It was noted that this finding is drawn from analyses utilizing a 90% confidence interval. This confidence level indicates that while minor discrepancies may exist in the data, they are considered negligible under specific statistical assumptions. Therefore, we conclude that random sampling provides a practical and statistically robust alternative, particularly in scenarios where it is not feasible to retain the characteristics of diurnal cycles or seasonality.

### 4.4 Limitations of this study

While our study focuses on long-term wind data from five coastal onshore stations in Norway, it may not fully represent offshore wind conditions. Although these stations are all located at low elevations and near the coastline, their degree of exposure to open-sea winds varies due to local topography, coastal geometry, and sheltering effects (Fig. S15). For example,

SN35860 and SN44080 are directly exposed to the open sea, while SN38140 is partially sheltered by inland terrain and surrounding vegetation. Offshore wind can differ significantly from those onshore. In our study, ERA5 data tends to overestimate the frequency of high wind events at coastal sites. By contrast, recent study indicates that ERA5 may underestimate strong wind speed offshore (Gandoin and Garza, 2024), suggesting that discrepancies may stem from differences in surface roughness, atmospheric stability, and model representation of marine boundary layers. This highlights the need for

targeted offshore studies, for example using buoy-based wind measurements (Morgan et al., 2011). Furthermore, our analysis does not include complex inland terrains such as mountainous regions or deep valleys, where wind speed distributions can be bimodal (Jaramillo and Borja, 2004) or strongly affected by topographic channelling. These environments are likely to show different sensitivities to sampling strategies, especially about shape-related distribution metrics. We therefore recommend that future research apply this framework to both offshore locations and inland complex terrain to better capture the full range of

wind resource variability and distributional stability.

Moreover, we compared the surface elevation of the ERA5 grid cells with the actual heights of the five Norwegian weather stations (Table 1). While all stations are situated near sea level (ranging from 4 m to 48 m above mean sea level), ERA5 grid elevations differ substantially, with four out of five stations showing discrepancies exceeding 40 m, and one exceeding 110 m.

Specifically, ERA5 overestimates elevation at three stations and underestimates it at two. Interestingly, despite the mix of elevation biases, ERA5 wind speeds are overestimated at four stations and underestimated at only one. A station where ERA5 overestimated elevation is also the one where wind speed is underestimated. This suggests that elevation mismatch alone cannot fully explain the direction or magnitude of wind speed biases. Other factors, such as surface roughness and land use type, may also contribute to the discrepancies.

Furthermore, our analysis does not include complex inland terrains such as mountainous regions or deep valleys, where wind speed distributions can be bimodal (Jaramillo and Borja, 2004) or strongly affected by topographic channelling. These environments are likely to show different sensitivities to sampling strategies, especially about shape-related distribution metrics. We therefore recommend that future research apply this framework to both offshore locations and inland complex

terrain to better capture the full range of wind resource variability and distributional stability.

Another limitation is the time resolution of the wind speed data we used. We utilized hourly data instead of higher temporal resolution data, such as 10-minute intervals, for wind distribution assessments. Despite this, Yang et al., (2024) demonstrated

that hourly wind speed data provide sufficiently accurate estimations of wind power density, with errors smaller than ±2%
when compared to 10-minute resolution data. This suggests that hourly data are suitable for such analyses. Additionally, Effenberger et al., (2024) showed that three- or six-hourly instantaneous wind speed data can effectively preserve the distribution characteristics of 10-minute wind speeds. Therefore, it is reasonable that hourly wind speed can adequately represent the characteristics of 10minute wind speeds.

It is worth noting that the hourly data provided by MET Norway represent the average wind speed over the last ten minutes of each hour rather than the entire hour. Despite this, previous research found that Weibull distribution parameters remain consistent across different averaging periods (e.g., 1 minute and 30 minutes) (Barthelmie and Pryor, 2003). Based on these findings, we believe that our use of last 10-minute averages is unlikely to significantly impact the accuracy of the Weibull distribution parameters compared to full-hour averages.

Additionally, our study focuses on near-surface wind speeds (10 m), raising questions about whether our conclusions hold at turbine-height winds. Prior studies indicate a height dependency for Weibull distribution parameters, with higher altitudes typically showing higher means (and scale parameter), variances, skewness, and kurtosis, while the shape parameter remains height-independent (Barthelmie and Pryor, 2003; Dixon and Swift, 1984). Due to the absence of observational data at heights other than 10 meters, we utilized the ERA5 dataset to compare distribution parameters at 10-m and 100-m heights. For the five locations studied, only the mean (and Weibull scale parameter), and variance show height dependency, with other parameters (skewness, kurtosis, Weibull shape parameter) showing independence from height.

**5 Conclusions**

Our study quantifies the errors in estimating wind speed distribution parameters using time series of varying lengths, accounting for interannual variability. We find that skewness and kurtosis, particularly kurtosis, are systematically underestimated when data length is limited, and this underestimation is more pronounced in datasets with higher skewness and kurtosis levels, necessitating significantly longer observation periods for accurate estimates. While the mean and standard deviation stabilize within weeks of data, skewness requires over 1.6 years and kurtosis over 88.8 years for a ±5% error margin. These results emphasize that the required length of wind observations is strongly dependent on the shape characteristics of the underlying distribution, with regional variations becoming more pronounced as accuracy demands increase, particularly for higher-order statistical properties like skewness and kurtosis.

These findings have important implications for wind resource assessment, particularly in regions characterized by highly variable wind regimes. In such areas, extended data collection periods or alternative strategies such as data fusion or machine

learning may be essential to accurately capture higher-order statistical properties, which directly affects energy yield estimates and turbine design standards.

We also compare different sampling strategies. Our results show that random sampling yield more statistically efficient estimates than continuous sampling, which preserves temporal correlation and diurnal pattern but introduces greater variability in estimated parameters. For instance, achieving ±10% uncertainty in power density may require at least five years of continuous data, whereas only about two months of randomly sampled hourly data may suffice. This suggests that flexible sampling approaches may be feasible in data-limited environments, provided the sampling design avoids strong temporal clustering.

Finally, our evaluation of ERA5 reanalysis data reveals that although such datasets require fewer data points for the same error margin, they introduce systematic biases, such as underestimating skewness and overestimating Weibull shape parameters, compared to in-situ measurements. This underscores the need for caution when using reanalysis data in wind resource assessments, particularly in regions with complex wind regimes.

Future studies should focus on mitigating biases in higher-order moment estimation. Moreover, extending this analysis to different terrain types, and hub   heights can further improve the reliability and generalizability of wind energy assessments.

**Code availability**

The code used in this paper can be obtained from the author upon request.

**Data availability**

The observed wind speed dataset from MET Norway is available for download from at MET Norway's FROST platform (https://frost.met.no/index.html; last access: 8 February 2025). The ERA5 datasets is available at Copernicus Climate Data Store (https://cds.climate.copernicus.eu/datasets/reanalysis-era5-single-levels?tab=download; last accessed 8 February 2025).

**Author contributions**

LH conceptualized the article, wrote it, and conducted the analysis, while IE supervised the project and contributed to the interpretation of the results and the writing.

**Competing interests**

The authors declare that they have no conflict of interest.

**Acknowledgements**

The work is part of the project "UiT - Intermittent character of wind energy resources on different spatial and temporal scales",
funded by the Faculty of Science and Technology, University of Tromsø. Igor Esau acknowledges a contribution from the ESA project MAXSS 4000132954/20/I-NB.

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
