# Peer review of "Determining the ideal length of wind speed series for wind speed distribution and resource assessment"

_Wind Energy Science, 2025_

## Author Response (AR1)

**Authors' Response to Reviewer 1**

*Reviewer's Comment #1*: I am not completely sure of your distinction between good and fair in the grading. I have indicated the level "fair" in the meaning the paper meets the expected requirements for a scientific paper.

***Response to Comment #1***: Thank you very much for the clarification. We appreciate your constructive feedback and have carefully revised the manuscript based on your comments to improve both the clarity and relevance of the work. We hope the revised version will better reflect the strengths of the study.

*Reviewer's Comment #2*: My comments regarding the revisions: The authors have quantified the levels of accuracy of using wind speed series of different lengths regarding estimates of various statistical parameters related to wind energy. The results are however - as they also state - limited to Southern Norway using coastal weather stations only. The authors invite more studies for other areas. Although not a request, it would have added extra value if it somehow had been included in this paper.

***Response to Comment #2***: Thank you for your thoughtful and constructive suggestion. We have extended our analysis by incorporating two additional coastal stations located outside Norway—one in Denmark and one in Scotland. These sites are situated in or near operational wind farms in the North Sea and thus serve as relevant and practical complements to our original Norwegian stations.

Both stations provide 46 years of hourly wind speed observations, allowing us to test the robustness of our findings in different geographical and climatological settings. Specifically, we used these longer and uninterrupted datasets to explore the differences between random and continuous sampling approaches, which also responds to related concerns in Comment #3. This additional analysis enhances the generalizability of our guidance on data length requirements for wind resource assessments.

To maintain the coherence of the manuscript structure, we have incorporated this complementary analysis into Section 4.1 of the Discussion. We also present the added contents as follows:

[Figure]

"**Figure 1: Distribution of the weather stations used in this study.** (*Line 205 in the clean version of the revised manuscript*)"

"*4.1 Sensitivity to sampling strategy and climatic non-stationarity*

*In wind energy assessments, continuous sampling is more commonly used than random sampling because it preserves temporal structure and seasonal variability in wind speed time series, and most importantly, only long-term data are not available. However, continuous sampling may also introduce systematic bias, particularly over short durations, due to temporal autocorrelation and underlying climatic non-stationarity. To investigate the extent of this effect and assess the generalizability of random sampling, we conducted a sensitivity analysis using 46 years (1979–2024) of hourly wind speed data from two coastal meteorological stations: Copenhagen Airport (061800-99999, Denmark) and Leuchars (031710-99999, Scotland). These sites were chosen for their long-term records and meteorological similarity to the five Norwegian locations analyzed earlier. Copenhagen station exhibits a long-term decreasing wind speed trend (Fig. S1), consistent with broader global observations (Zeng et al., 2019).*

*Our results show that continuous sampling generally requires significantly longer periods to achieve the same level of uncertainty in estimated distribution parameters compared to random sampling (Fig. 7). This discrepancy arises because random sampling draws from multiple years, thereby capturing a wider range of interannual variability and reducing exposure to temporal clustering. Consequently, the 90% confidence intervals (CIs) under random sampling are symmetric for all parameters, while under continuous sampling, only the CIs for mean wind speed, Weibull scale parameter, and power density are symmetric. Shape-sensitive parameters, including standard deviation, skewness, kurtosis, and especially the Weibull shape parameter, exhibit pronounced asymmetries under continuous sampling, particularly at short durations (<2 years). This suggests that the presence of systematic climatic anomalies in continuous subsets may bias shape estimation.*

*These findings support earlier recommendations by Murthy et al. (2017), who advocate using at least four to ten years of data for reliable wind energy assessments. Our results suggest that when using continuous sampling, at least five years of data may be required to achieve ±10% relative uncertainty in power density*

*estimates, although this threshold is site-specific (e.g., Copenhagen station requires more than 10 years).
We further recommend that random sampling be considered as a complementary tool to identify potential
biases in short-term continuous assessments.*" (*Lines 355-379 in the clean version of the revised
manuscript*)

[Figure]

**Figure 7: distribution parameters and Weibull power density derived from random sampling (orange lines) and continuous sampling (black lines), based on in-situ measurements from weather stations.** Asterisks indicate values computed from the full 46-year dataset. Values for sample lengths between 14 and 46 years are omitted for

visual clarity. Details of the experimental setup and sampling procedures are provided in the Methods section. (*Lines 380-385 in the clean version of the revised manuscript*)

*Reviewer's Comment #3*: Given that the paper is focusing on wind energy it is a bit surprising that the Wind Atlas approach is not discussed in the paper. Not even mentioned. I would like to see a discussion of wind atlas and the wind energy intensity estimates.

*Response to Comment #3*: Thank you for highlighting this important point. In response, we have added a dedicated subsection (Section 4.2) in the Discussion, which introduces the Wind Atlas methodology and discusses its relevance to wind energy resource assessments. This section also presents a comparison between Global Wind Atlas estimates and our station-based observations in terms of key wind energy metrics. We believe this addition enhances the energy-focused perspective of the paper and addresses your suggestion effectively.

"*4.2 Evaluation of global wind atlas estimates against observations*

*Since the publication of the first European Wind Atlas in 1989 (Dörenkämper et al., 2020), the wind atlas methodology has been widely adopted for regional wind resource assessments, including in countries such as Finland (Tammelin et al., 2013) and Greece (Kotroni et al., 2014). The Global Wind Atlas (GWA), developed by the Technical University of Denmark, applies the well-established numerical wind atlas method to downscale coarse-resolution reanalysis data to microscale levels. This is achieved using linearized flow models and topographic corrections based on the WAsP model. GWA provides publicly accessible estimates of mean wind speed and power density, which have been used in applications such as bias correction of reanalysis data for wind power simulations (Gruber et al., 2022).*

*Given the energy-focused perspective of this study, it is relevant to compare our results with GWA estimates. We extracted GWA values at the nearest grid points for selected stations and compared them with observational estimates based on the full time series. Table S7 presents this comparison, focusing on two key metrics in wind energy assessments: mean wind speed and power density. The results show that GWA consistently overestimates both wind speed and power density relative to our station-based observations.*

*One likely explanation for this discrepancy lies in the different ways topographic effects are incorporated. As described by Davis et al. (2023), the GWA estimates the predicted wind climate (PWC) by applying high-resolution topographic perturbations to the generalized wind climate which is based on coarse reanalysis fields. The PWC is represented by a set of Weibull distributions and directional frequencies for each of 12 directional sectors, and these are used to calculate derived variables such as mean wind speed and power density.*" (*Lines 412-433 in the clean version of the revised manuscript*).

*Reviewer's Comment #4*: They compare their analysis of observed wind speed with an equivalent analysis of reanalysis data using ECMWF data, ERA5.

They don't discuss the topography issue. This is important, as one must expect the height of the surface in the nearest gridpoint to be crucial to the interpretation of reanalyzed wind fields very near the surface or close to the surface.

***Response to Comment #4***: Thank you for highlighting this important point. In response, we added a new paragraph to the revised manuscript to discuss the surface elevation differences between the ERA5 grid cells and the actual station locations. The elevation values are now also included in Table 1.

"*Moreover, we compared the surface elevation of the ERA5 grid cells with the actual heights of the five Norwegian weather stations (Table 1). While all stations are situated near sea level (ranging from 4 m to 48 m above mean sea level), ERA5 grid elevations differ substantially, with four out of five stations showing discrepancies exceeding 40 m, and one exceeding 110 m. Specifically, ERA5 overestimates elevation at three stations and underestimates it at two. Interestingly, despite the mix of elevation biases, ERA5 wind speeds are overestimated at four stations and underestimated at only one. A station where ERA5 overestimated elevation is also the one where wind speed is underestimated. This suggests that elevation mismatch alone cannot fully explain the direction or magnitude of wind speed biases. Other factors, such as surface roughness and land use type, may also contribute to the discrepancies.*" (*Lines 462-469 in the clean version of the revised manuscript*).

**Table 1: Details of weather stations used in this study.** (*Lines 201-203 in the clean version of the revised manuscript*).

| Station ID | Location | Data source | WMO number | Latitude | Latitude of ERA5 grid | Longitude | Longitude of ERA5 grid | Height above mean sea level | Elevation of ERA5 grid |
|---|---|---|---|---|---|---|---|---|---|
| SN50500 | Flesland | | 1311 | 60.2892° N | 60.25° | 5.2265° E | 5.25° | 48 m | 0.3 m |
| SN44080 | Obrestad Fyr | | 1412 | 58.6592° N | 58.75° | 5.5553° E | 5.50° | 24 m | 5.6 m |
| SN42160 | Lista Fyr | MET Norway | 1427 | 58.1090° N | 58.00° | 6.5675° E | 6.50° | 14 m | 127.1 m |
| SN38140 | Landvik | | 1464 | 58.3400° N | 58.25° | 8.5225° E | 8.50° | 6 m | 55.4 m |
| SN35860 | Lyngør Fyr | | 1467 | 58.6362° N | 58.75° | 9.1478° E | 9.25° | 4 m | 43.9 m |
| 061800-99999 | Kastrup | HadISD | / | 55.618° N | / | 12.656° E | / | 5.2 m | / |
| 031710-99999 | Leuchars | | / | 56.373° N | / | -2.868° E | / | 11.6 m | / |

Note: As the last two stations (Kastrup and Leuchars) were added specifically for the sensitivity analysis discussed in Section 4.1, they were excluded from the comparison with ERA5.

*Reviewer's Comment #5*: One of the objectives in the paper is addressing the possibility that one can use randomly (in the time series) selected data to obtain the necessary distribution parameters. They state that this method is working for both near the surface and at elevated levels. And they then use this method for their following analyses.

*Response to Comment #5*: Yes, addressing the feasibility of using randomly selected time samples to estimate wind speed distribution parameters is one of the main objectives of our study. This approach is motivated by the fact that many meteorological stations have long-term wind speed records that are incomplete or discontinuous in time. Traditionally, such datasets are excluded from wind resource assessments due to their temporal gaps. However, our results show that by applying random sampling to these fragmented but long-term records, it is still possible to capture key distribution characteristics.

In our analysis, we demonstrated the feasibility of this approach by comparing the 90% confidence intervals of distribution parameters obtained through different sampling methods. Furthermore, the revised manuscript now includes a comparison with continuous sampling. The results indicate that continuous sampling generally requires significantly longer data periods to achieve comparable uncertainty levels. A more detailed comparison between random and continuous sampling methods is also provided in our response to Comment #1.

We acknowledge, as stated in the manuscript (Lines 442–445), that these findings are based on analyses using a 90% confidence interval. This level implies that while minor discrepancies may exist, they are negligible under certain statistical assumptions. Therefore, we conclude that random sampling provides a practical and statistically robust alternative, particularly in situations where preserving diurnal or seasonal structures is not feasible.

"*It was noted that this finding is drawn from analyses utilizing a 90% confidence interval. This confidence level indicates that while minor discrepancies may exist in the data, they are considered negligible under specific statistical assumptions. Therefore, we conclude that random sampling provides a practical and statistically robust alternative, particularly in scenarios where it is not feasible to retain the characteristics of diurnal cycles or seasonality.*" (*Lines 442-445 in the clean version of the revised manuscript*).

*Reviewer's Comment #6*: Page 8 and page 10 contain very detailed graphical illustrations of this point. They are however not easily grasped: 5 stations and 7 parameters giving 35 small plots. Even in a A4 print it is not a simple exercise to see every point discussed. Especially in figure 3 the blue colors are not easily identified.

Page 10, figure 3, contain an extra column with a1) - a6) superimposed on the middle column of small plots. Should be removed.

***Response to Comment #6***: Thank you for your valuable feedback regarding the visual clarity of the figures. In response, we have revised the figures to improve readability. Specifically, we retained only three key variables, mean wind speed, Weibull scale parameter (*c*), and power density, for the five Norwegian stations in the main manuscript, and moved the remaining variables to the supplementary materials. In addition, we have adjusted the color scheme to enhance visual distinction, particularly addressing the issue raised about the blue tones in Figure 3. The superimposed column labels (a1)–a6)) previously shown in the middle column of Figure 3 have also been removed, as suggested. An example of the revised figure is shown below:

[Figure]

**Figure 2: Estimates of mean wind speed, Weibull scale parameter, and power density from three sampling strategies, based on in-situ observations from five Norwegian stations.** The 90% confidence intervals (CIs) are shown for each sampling method: random (orange), diurnal-cycle-retained (purple dashed), and seasonality-retained (blue dotted). Each black dot represents a parameter estimate from a single sampling realization of random sampling; corresponding realizations for the other two methods are not shown. Sample sizes range from 720 to 52,560 (30 days to 6 years), increasing in 240-hour (10-day) increments, with 1,000 realizations per size. Red asterisks indicate the reference values from the full 16-year hourly dataset (see Table 2). Shaded areas represent ±2% (dark blue) and ±5% (light blue) deviation ranges from full-series values. (*Lines 220-228 in the clean version of the manuscript*)

***Reviewer's Comment #7***: The authors distinguish between statistical parameters that are quickly obtained, like the mean, st.dev. and Weibull parameters, and other parameters like skewness and kurtosis requiring much longer time, respectively 1,6 years and 88 years of data.

I cannot easily see where these numbers are coming from.

***Response to Comment #7***: Thank you for pointing this out. The values of 1.6 years for skewness and 88 years for kurtosis refer to the estimated amount of hourly wind observations required to achieve a given level of accuracy in estimating these shape-related parameters. These estimates correspond to the least demanding stations in our study: for skewness, 14,084 hourly observations are required at station SN35860, which is equivalent to approximately 1.6 years of data. For kurtosis, 777,573 hourly observations are required at station SN38140, corresponding to about 88 years. These values are reported in Table 4 of the revised manuscript. We have clarified the process of how these values were obtained and revised the corresponding explanation in the manuscript to improve transparency. The updated text is shown below:

"*3.3 Determine an effective sample size for capturing overall wind characteristics*

*To determine the optimal sample size for capturing wind characteristics, we analysed the relationship between percent errors and sample sizes (Fig. 4-5). Percent error measures discrepancies between parameters from the full dataset and smaller subsets. Based on the 90% CIs derived from 1,000 realizations of random sampling of in-situ observations (orange lines in Fig. 2 & Fig. S4), we computed percent errors of CI bounds and fitted power-law equations to describe their dependence on sample size. These fitted equations are summarized in Table 3 and allow extrapolation of error margins for any given sample size.*" (Lines 272-277 in the clean version of revised manuscript)

"*To facilitate practical use, we calculated the minimum sample sizes required to achieve ±10%, ±5%, ±2%, and ±1% error margins for each parameter at each station (Table 4).*" (Lines 286-287 in the clean version of revised manuscript)

***Reviewer's Comment #8***: A statement about a 88 year time scale based on a time series not longer than 16 years is a bit strange. How is this calculation done? It must be based on some assumptions, but which ones? And what about non-stationarity features of the time series like effects of climate changes?

***Response to Comment #8***: The 88-year time scale mentioned in the manuscript is not derived from actual observational data of such duration but rather estimated using the fitted equations that describe the relationship between percent error and sample size, as shown in Table 3 of the manuscript. These equations were obtained through curve fitting based on the random sampling results, as described in detail in the response to Comment #7.

Once the equations are established, they allow us to estimate the sample size (in hours or years) required to achieve a given level of percent error. For instance, if one sets a percent error threshold, the model may suggest that up to 88 years of continuous data would be required to meet it. This is a theoretical extrapolation

Regarding the reviewer's concern about non-stationarity due to climate change, we fully agree this is an important issue. However, for the purposes of this study, we assume stationarity. Our aim is to assess how the length of the dataset affects the estimation of wind distribution parameters, not to assess trends in the wind climate. The reference long-term datasets are used to characterize the current wind climate baseline, not to make future projections. We acknowledge that climate-driven changes in wind characteristics could influence the applicability of these estimates in future conditions, and we agree this is a valuable direction for future work.

**Authors' Response to Reviewer 2**

*Reviewer's Comment #1*: In my view, the paper should be subject to major revision.

The study sets out an objective of very high practical importance for wind resource analysis: investigate whether short-term wind speed data from WMO weather stations realistically represent the wind speed statistics. Observations frequently cover a limited time span, which may introduce substantial errors in the estimation of several important parameters of the wind distribution, like its time variability. Therefore, guidance for selecting adequate datasets of adequate time spans is highly relevant.

*Response to Comment #1*: We appreciate the recognition of the practical significance of our study. We sincerely thank the reviewer for the thoughtful and constructive evaluation. We acknowledge the recommendation for a major revision, and in response, we have conducted a comprehensive revision of the manuscript. Specifically, we have addressed all specific reviewer comments in detail; improved the clarity and transparency of our methodology and assumptions; added new comparative analyses (e.g., with continuous sampling and Wind Atlas data) to strengthen the generality and relevance of our findings. We hope these revisions address the reviewer's concerns and contribute to a significantly improved manuscript.

*Reviewer's Comment #2*: In my view, the applied method is not adequate for this aim ("provide guidance for selecting adequate datasets of adequate time spans"). From the introduction and the description of the objectives of the study, it seems that datasets with different lenghts (720 hours to 6 years) would correspond to series of consecutive hours, so that a 720 hours dataset would correspond to 1 month of consecutive hourly data. In this way, actual observations of different lenghts would be replicated.

But from the description of the method (random sampling) it seems that the authors randomly select 720 separate hours within the full 16-year time series. Random sampling over a long time span of 16 years has much less practical relevance, due to the fact that actual station observations will be based on continuous (or nearly continuous) datasets, not on random measurements with random and large gaps between them. The small data gaps typically found in useful observations do not change this, as the full time span of the observations will still be limited.

For example, a continuous (or nearly continuous) dataset of a few months will not cover adequately the seasonal variability of the wind, but if the data are randomly selected over a large period the seasonal variability may be well represented, as indicated by the results of the study. From a practical point of view, a 720 hourly data dataset randomly selected from a multiyear full time series is not equivalent to a continuous (or nearly continuous) 720-hour dataset covering one month in total.

The result of the paper indicating that mean, standard deviation, and Weibull parameters, stabilize with relatively short records (~1 month of randomly selected hourly data) may not hold for 1-month continuous data.

The study builds on the work of Barthelmie and Pryor (2003), but that paper has a fundamental difference, as their sampling is not random. They use conditional samples to replicate data that could be obtained from remote sensing tools, which gives that paper a high practical relevance.

*Response to Comment #2*: We thank the reviewer for this insightful comment, which has helped clarify the interpretation of our methodology and its relation to our stated objectives. Our response is structured in two parts. First, we address the reviewer's remark that "Barthelmie and Pryor (2003) has a fundamental difference, as their sampling is not random." We would like to clarify that our random sampling method follows the same approach as in Barthelmie and Pryor (2003). The only technical difference lies in the choice of random number generator—the Park and Miller "Minimal Standard" in their study, versus a Permuted Congruential Generator in ours. As stated in their paper:

"*To examine the dependence of the distribution parameters on dataset density (i.e., number of observations in the time series) the dataset from Vindeby SMW was randomly and multiply resampled for a range of number of observations from n = 21 (assumed to be the lower bound on the dataset likely to be obtained using remote sensing) to n ~ 0.1 of the actual number of observations available from Vindeby for the entire data collection period (i.e., n = 10 000). The resampling was undertaken with sample replacement (so the same observation could be selected multiple times within one resampling group and/or in two or more of the 1000 resampling groups for each n) using the Park and Miller "Minimal Standard" random number generator. The results are presented in Fig. 2 for the four moments of the distribution and the Weibull parameters for 1000 resampling iterations for each n.*"

They further noted that their method did not retain seasonality or diurnal cycles, which motivated us to develop additional sampling strategies (i.e., diurnal-cycle-retained and seasonal-cycle-retained sampling) to investigate the effects of such temporal structures. They also acknowledged this limitation: "*It should be noted that a critical aspect of the applicability of the results (and uncertainty bounds) presented here is that the datasets are randomly drawn from the time series with respect to seasonality.*".

Second, regarding the reviewer's concern that random sampling does not reflect the practical reality of continuous datasets used in operational or observational settings, we fully agree. In response, and as elaborated in Comment #3, we incorporated additional analyses based on continuous sampling, using two stations with 46 years of uninterrupted hourly wind data. This allows for a direct comparison between random and continuous sampling, addressing the core

concern raised here. We believe this addition substantially improves the practical relevance of the study and provides the guidance originally intended in our objectives.

*Reviewer's Comment #3*: If there is some misunderstanding of the method on my side, please clarify the description of the method. If I have correctly understood the method, I would suggest two possibilities:

1. - Change the objectives of the study to adapt them to the method. The practical relevance of the study would be substantially smaller, but it is in any case an interesting investigation about the characteristics of wind distributions. The comparison of random sampling to diurnal-cycle-retained and seasonality-retained data is really interesting in itself. Several aspects of the study offer new insights on wind distributions, like the different error margins analysed, and the comparative analysis of ERA data.

2. - Change the method to adapt it to the objectives. This could be done by selecting for example many different continuous datasets of the same length (e.g. 720 hour) within the full 16-year time series. The random sampling calculations already done could be retained for comparison purposes. A possibility to take advantage of the work already done, which is technically well performed, would be to use the results from the random sampling calculations to select the lenghts of the continuous datasets to analyse. That is, not all lengths until 6 years would be analysed with continuous datasets, only lengths from 720 hours to a threshold determined from the errors obtained in the random sampling calculations.

*Response to Comment #3*: We sincerely thank the reviewer for the thoughtful and constructive suggestions. In response, we followed the second suggestion proposed by the reviewer, namely, adapting the method to better align with the original objectives, by incorporating additional analyses based on continuous sampling.

To implement this, we used two additional stations located in Denmark and Scotland, each with 46 years of hourly wind speed observations, which allowed for sufficiently long and uninterrupted time series required for continuous sampling. The five Norwegian stations originally used in the study contain substantial data gaps and were therefore unsuitable for this type of analysis. Another reason for selecting these two additional stations is their geographical relevance: both are situated along the coast in the North Sea region, which is consistent with the coastal characteristics of the Norwegian stations.

The results of this additional analysis indicate that continuous sampling generally requires significantly longer time periods to achieve the same level of uncertainty in estimated distribution parameters, compared to random sampling. These findings reinforce the practical utility of random sampling when long, uninterrupted records are not available. Details of the comparison between

random and continuous sampling have been added to the Discussion section of the revised manuscript. We also show the complementary contents below:

"*4.1 Sensitivity to sampling strategy and climatic non-stationarity*

*In wind energy assessments, continuous sampling is more commonly used than random sampling because it preserves temporal structure and seasonal variability in wind speed time series, and most importantly, only long-term data are not available. However, continuous sampling may also introduce systematic bias, particularly over short durations, due to temporal autocorrelation and underlying climatic non-stationarity. To investigate the extent of this effect and assess the generalizability of random sampling, we conducted a sensitivity analysis using 46 years (1979–2024) of hourly wind speed data from two coastal meteorological stations: Copenhagen Airport (061800-99999, Denmark) and Leuchars (031710-99999, Scotland). These sites were chosen for their long-term records and meteorological similarity to the five Norwegian locations analyzed earlier. Copenhagen station exhibits a long-term decreasing wind speed trend (Fig. S1), consistent with broader global observations (Zeng et al., 2019).*

*Our results show that continuous sampling generally requires significantly longer periods to achieve the same level of uncertainty in estimated distribution parameters compared to random sampling (Fig. 7). This discrepancy arises because random sampling draws from multiple years, thereby capturing a wider range of interannual variability and reducing exposure to temporal clustering. Consequently, the 90% confidence intervals (CIs) under random sampling are symmetric for all parameters, while under continuous sampling, only the CIs for mean wind speed, Weibull scale parameter, and power density are symmetric. Shape-sensitive parameters, including standard deviation, skewness, kurtosis, and especially the Weibull shape parameter, exhibit pronounced asymmetries under continuous sampling, particularly at short durations (<2 years). This suggests that the presence of systematic climatic anomalies in continuous subsets may bias shape estimation.*

*These findings support earlier recommendations by Murthy et al. (2017), who advocate using at least four to ten years of data for reliable wind energy assessments. Our results suggest that when using continuous sampling, at least five years of data may be required to achieve ±10% relative uncertainty in power density estimates, although this threshold is site-specific (e.g., Copenhagen station requires more than 10 years). We further recommend that random sampling be considered as a complementary tool to identify potential biases in short-term continuous assessments.*" (*Lines 355-385 in the clean version of the revised manuscript*)

[Figure]

**Figure 7: distribution parameters and Weibull power density derived from random sampling (orange lines) and continuous sampling (black lines), based on in-situ measurements from weather stations.** Asterisks indicate values computed from the full 46-year dataset. Values for sample lengths between 14 and 46 years are omitted for visual clarity. Details of the experimental setup and sampling procedures are provided in the Methods section.

*Reviewer's Comment #4*: A minor point that could be improved is the presentation of the results are the figures. There are too many frames in every figure that make it difficult to see the most relevant results. A selection of a couple of meteorological stations, that are representative of the whole set of stations, could perhaps be done. The full graphs could be moved to the supplementary information section. Also, the 3 curves limiting the 90% confidence intervals are difficult to distinguish, as they largely overlap.

*Response to Comment #4*: Thank you for this helpful suggestion. This issue was also raised by Reviewer #1, and we have made corresponding revisions in the manuscript. Specifically, we now present only three key variables—mean wind speed, Weibull scale parameter (*c*), and power density. The remaining variables have been moved to the Supplementary Materials to reduce visual clutter and enhance readability. We also revised the color scheme and line styles to improve the visual distinction of the 90% confidence interval curves, which previously overlapped and were difficult to differentiate. An example of the revised figure is provided below, as it appears in the revised manuscript.

[Figure]

**Figure 2: Estimates of mean wind speed, Weibull scale parameter, and power density from three sampling strategies, based on in-situ observations from five Norwegian stations.** The 90% confidence intervals (CIs) are shown for each sampling method: random (orange), diurnal-cycle-retained (purple dashed), and seasonality-retained (blue dotted). Each black dot represents a parameter estimate from a single sampling realization of random sampling; corresponding realizations for the other two methods are not shown. Sample sizes range from 720 to 52,560 (30 days to 6 years), increasing in 240-hour (10-day) increments, with 1,000 realizations per size. Red asterisks indicate the

reference values from the full 16-year hourly dataset (see Table 2). Shaded areas represent ±2% (dark blue) and ±5% (light blue) deviation ranges from full-series values. (*Lines 220-228 in the clean version of the manuscript*)

---

## Author Response (AR2)

**Authors' Response to Reviewer 2**

*Reviewer's Comment #1*: The changes and new additions have improved the paper, and the authors have sufficiently addressed my concerns, in general.

There is nevertheless still one aspect that should be improved, in my view. The number of discontinuous random data is quantified on several occasions as "months" or "years" of data, instead as sample size or number of data. This can cause confusions, as "1 month" or "1 year" of random data could be understood as the actual total duration of observation datasets. In reality, "1 month" of random data requires many years of observations in order to get the favorable results shown in the paper.

The clearest way to avoid this confusion would be to remove any mention of "months" or "years" of data when referring to the number of discontinuous random data. At least, a clear explanation of the specific meaning of these time periods should be made, and a wording like "equivalent in size to 1 month" could be used. The use of such time references is particularly misguided in the abstract, as the procedure (random sampling of multi-year datasets) for obtaining the result that "basic parameters, such as mean, standard deviation, and Weibull parameters, can stabilize with ~1 month of hourly data" has not been explained before this statement. Also, in figure 7 the x-axes is written as "Number of data (yrs)", whereas in figures 2 to 6 the actual number of data is used in the x-axes. Using also just "Number of data" in figure 7, as in previous figures, would be clearer. There are several other sentences throughout the text where the use of "months" or "years" should be cancelled or clarified.

*Response to Comment #1*: We agree that referring to the size of discontinuous random samples as "months" or "years" can be confusing. We carefully revised the manuscript to remove ambiguous uses when referring to discontinuous random samples by doing these revisions:

- Include the original hourly count and use an "years-equivalent" note to denote sample size in time units, for example:

Abstract: "*We apply this method to in-situ station observations and ERA5 reanalysis data at 10 m and 100 m heights. Our results show that basic parameters (mean, standard deviation, and Weibull parameters) stabilize with a sample size equivalent to ~1 month of hourly data (not a contiguous period) drawn across multiple years, while higher-order moments require substantially larger samples (skewness: ~1.6 years equivalent; kurtosis: 88.6 years equivalent). Although ERA5 stabilizes faster, it exhibits systematic biases compared to in-situ measurements. Moreover, random cross-year sampling yields comparable distribution parameters to diurnally or seasonally controlled sampling, while continuous sampling demands far longer records for the same accuracy.*" (*Lines 13-19 in the clean version of the revised manuscript*)

"*The kurtosis bias remained above 10% until sample size exceeded 2 160 hours, and SN50500 required 22 080 observations (~2.5 yrs equivalent) to reduce error to within 10%.*" (*Lines 268-269 in the clean version of the revised manuscript*)

"*For example, ±5% accuracy requires 459 hourly observations for the mean, 470 for the Weibull scale (~20 days equivalent), 796 for standard deviation (~34 days equivalent), and 4 031 for power density. Achieving ±2% and ±1% error requires 6-fold and 24-fold of observations than ±5% case, respectively. Skewness and kurtosis are especially data-intensive due to their sensitivity to distribution tails. For instance, SN38140 needs 177 390 observations (~20 years equivalent) for ±10% error, while SN50500 needs 1 541 437 observations (~176 years equivalent).*" (*Lines 286-291 in the clean version of the revised manuscript*)

Conclusion: "*While the mean and standard deviation stabilize with a few hundred hourly samples, skewness requires at least 14 084 hours and kurtosis at least 777 573 hours to meet a ±5% error margin (1.6 years and 88.6 years-equivalent, respectively). Here, "years-equivalent" denotes the number of hourly observations equal to the hours in that duration and does not imply a contiguous period (samples are randomly drawn across years). These results emphasize that the required sample size is strongly dependent on the shape of the underlying distribution, with regional differences becoming more pronounced as accuracy demands increase, particularly for higher-order statistical moments like skewness and kurtosis.*" (*Lines 497-503 in the clean version of the revised manuscript*)

- We revised the captions for Figures 4-5 and for the corresponding SI figures. We also updated Figure 7 to adopt the same x-axis convention as Figures 2-6 and added a secondary top axis showing the equivalent years.

"**Figure 4: The relationship between the percent error (Y) and sample size (n) (number of hourly observations) across five stations.** Curves are fitted for n = 720 ~ 140 160, with n = 720 is equivalent in size to 30 days of hourly data and 140 160 equivalent to 16 years. The equations of fits here are shown in Table 3. Grey circles indicate the values used to fit the 90% confidence intervals for the percent error shown." (*Lines 307-310 in the clean version of the revised manuscript*)

"**Figure 5: Same as Fig. 4, but the hourly observations ranging from $n$ = 24 ~ 720 across five stations.** These intervals are calculated using the same fits as shown in Fig. 4." (*Lines 312-314 in the clean version of the revised manuscript*)

[Figure]

**Figure 7: Distribution parameters and Weibull power density derived from random sampling (orange lines) and continuous sampling (black lines), based on in-situ measurements from weather stations.** The x-axis shows the number of hourly observations; a secondary top axis indicates the equivalent number of years (1 year = 8760 h). Asterisks indicate values computed from the full 46-year dataset. Details of the experimental setup and sampling procedures are provided in the Methods section. (*Lines 381-386 in the clean version of the revised manuscript*)

*Reviewer's Comment #2*: On the other hand, I suggest that the authors include an additional application field of the results: the analysis of long-term high-resolution climate simulations for wind energy assessments. The fact that a relatively small number of random data from long-period wind series can reproduce relevant characteristics of the wind distributions could be very helpful for managing the huge amount of data from long-term high-resolution climate simulations.

*Response to Comment #2*: We thank the reviewer for this valuable suggestion. It highlights an important additional application for our results and further underscores their relevance. In the Conclusion, we have added the following concise statement to make this use case explicit (we did not modify the Abstract due to word limits):

"*An additional application of this result is to long-term high-resolution climate simulations: rather than processing the full, continuous multi-decadal time series, a relatively small, randomly sampled set of hourly outputs spanning multiple years can recover the key wind-distribution characteristics. The required sample size can be determined from our sample-size-uncertainty relationships to meet a prescribed accuracy bound, while model biases and non-stationarity should be addressed separately.* " (*Lines 515-519 in the clean version of the revised manuscript*)

*Reviewer's Comment #3*: Typos: L55: change "if we must REPLY on short-term dataset" to "if we must RELY on short-term datasets"

*Response to Comment #3*: We have revised L55 accordingly: "*if we must rely on short-term datasets*" (*Line 51 in the clean version of the revised manuscript*)